# Effect of Polythiophene Content on Thermomechanical Properties of Electroconductive Composites

**DOI:** 10.3390/molecules26092476

**Published:** 2021-04-23

**Authors:** Katarzyna Bednarczyk, Tomasz Kukulski, Ryszard Fryczkowski, Ewa Schab-Balcerzak, Marcin Libera

**Affiliations:** 1Institute of Chemistry, University of Silesia in Katowice, 9 Szkolna Str., 40-006 Katowice, Poland; kbednarczyk@us.edu.pl (K.B.); ewa.schab-balcerzak@us.edu.pl (E.S.-B.); 2Faculty of Materials, Civil and Environmental Engineering, University of Bielsko-Biala, Willowa 2 Str., 43-309 Bielsko-Biala, Poland; tkukulski@ath.bielsko.pl (T.K.); rfryczkowski@ath.bielsko.pl (R.F.); 3Centre of Polymer and Carbon Materials, Polish Academy of Sciences, 34 M. Curie-Sklodowskej Str., 41-819 Zabrze, Poland

**Keywords:** composites, electrospinning, conducting polymer, polythiophene, morphology, thermomechanical analysis

## Abstract

The thermal, mechanical and electrical properties of polymeric composites combined using polythiophene (PT) dopped by FeCl_3_ and polyamide 6 (PA), in the aspect of conductive constructive elements for organic solar cells, depend on the molecular structure and morphology of materials as well as the method of preparing the species. This study was focused on disclosing the impact of the polythiophene content on properties of electrospun fibers. The elements for investigation were prepared using electrospinning applying two substrates. The study revealed the impact of the substrate on the conductive properties of composites. In this study composites exhibited good thermal stability, with T_5_ values in the range of 230–268 °C that increased with increasing PT content. The prepared composites exhibited comparable PA T_g_ values, which indicates their suitability for processing. Instrumental analysis of polymers and composites was carried out using Fourier Transform Infrared spectroscopy (FT-IR), thermogravimetric analysis (TGA), differential scanning calorimetry (DSC), dynamic mechanical thermal analysis (DMTA) and scanning electron microscopy (SEM).

## 1. Introduction

Electroconductive composites, due to their ability to form elastic constructive materials, are of major scientific and research interest with respect to their utilization for innovative solar cells, detectors or diodes [1,2]. Electrospinning of such composites an area of investigation due to the possible implication to produce elements such as photonic devices, among others [3,4,5,6,7].

Third-generation photovoltaics is one of the many areas of application for PT composite nanofibers. This polymer is an attractive alternative to the platinum counter electrode for dye-sensitized solar cells (DSSCs) due to its low cost, easy production, environmental friendliness, temperature stability and relatively high conversion efficiency. To date, many DSSCs in which the counter electrode was made of PT in different forms have been investigated [8].

The thiophene monomer can be polymerized by chemical or electrochemical methods [9]. First, polythiophene is generated by a metal-catalyzed polycondensation of 2,5-dibromothiophene [10]. Such a method leads to insoluble oligomers of molar mass below 3000 g/mol. One of the most often applied polymerizations of unsubstituted thiophene is synthesis with FeCl3 conducted in chloroform (Figure 1) [11], as also termed “classic”. The desirability of such a synthesis method is its unreproductivity, meaning that the same conditions of polymerization can result in different polymers (molar masses and its dispersion). Despite the limited processability of polythiophene made by the “classic” method of synthesis, they are still exploited to generate material for investigation of composites and films, among others.

Standard DSSCs consist of a working electrode (WE/photoanode), liquid electrolyte and counter electrode (CE/photocathode). As an alternative to conventional electrode materials, conductive nanocomposites composed of conducting nanomaterials and elastomeric media have been extensively studied for use in stretchable interconnections and devices. These stretchable conductive nanocomposites consist of percolation networks of nanoscale conductive fillers in elastomeric matrixes.

The usefulness of composite materials for advanced applications such as third-generation photovoltaics depends on the morphology and mechanical properties of the prepared elements, among others.

The formation of materials made of composites of PT in the form of fibers and nanofibers can lead to more efficient solar cells [12] and is essential in the production of flexible and wearable devices [13]. In particular, the development of PT nanofibers has significant benefits in comparison with film-casting polymers, as the former technique leads to a highly active surface area.

Electrospinning is one of the preparation methods for ultrathin fibers that have much smaller diameters than those obtained with conventional techniques [14]. Furthermore, by electrospinning, it is possible to adjust the properties of materials (e.g., fibers alignment, porosity and inner fiber structuration) [15]. The strong stretching forces and the physical confinement into the cylindrical fibers generated by electrospinning strongly affect the orientation of polymer chains along the long axis of a fiber [16]. Additional consequences of electrospinning of composites can be distinguished, taking into consideration the base layer applied for processing. Therefore, processing PT composites using electrospinning generates materials with new features [17].

The viscoelastic, thermal, morphological and conductive properties of materials induced by content of fillers in fibers, which creates thin mats made by electrospinning of conductive composites, can be revealed by correlative investigations of thermogravimetric, differential scanning, calorimetric and dynamic mechanical thermal analyses together with light scattering as well as electron microscopic observation. The storage modulus (E′) and the loss modulus (E″) are of major interest to determine, respectively, the elastic, viscous and damping properties of materials as well as the relationship of such properties with the structure and morphology of composite materials.

The present study focused on disclosing the impact of PT particles having a defined shape factor on the properties of composites composed of PT and polyamide (PA6). Herein, the base layer applied for electrospinning of mats will also be presented to expose the importance of such a factor. The results show that the properties of composite mats made of the same materials depend strongly not only on the content of fillers but also on the base layer applied for preparation, which should be taken under consideration during utilitarian functions.

## 2. Results

The syntheses of polythiophene in chloroform using ferric chloride as an oxidant yielded dark-brown powders. A scheme of oxidative polymerization is presented in Figure 1. After filtration, the powder was dried in a 60 °C oven under reduced pressure for 12 h and stored under an ambient atmosphere. The reaction yield after filtration was 84%.

Polythiophene was used to prepare composites with polyamide. The composites were prepared via electrospinning. Electrospinning conditions were selected based on the trial-and-error method. The composites were diversified by the polythiophene composite content and substrate type, and the data are collected in Table 1. The authors prepared materials by the electrospinning method, taking under consideration the flexibility and plasticity of the obtained composites. Electrospinning is known also as a method for improving the mechanical properties of composite films [18,19,20,21].

Polythiophene structure after synthesis was analyzed with FTIR studies at room temperature. The representative analysis of PAPT1Al is listed above. The results showed a characteristic absorption band at 1540 cm^−1^, which corresponds to C = C stretching vibration from the thiophene ring. C-C bonds were visible at 840 cm^−1^ and 1223 cm^−1^, which correspond to bending vibrations and stretching vibrations, respectively. The absorption peak at 758 cm^−1^ corresponded with stretching vibrations of the C-H bond. The C-S out-of-plane deformation bond vibrations were visible at 709 cm^−1^. The results confirmed the structure of polyaniline presented in the literature [22,23,24].

The thermal properties of polythiophene, polyamide and the thermomechanical properties of the manufactured composites were analyzed using TGA, DSC and DMTA measurements. Thermal stability was characterized by the temperatures at which 5% (T5) or 10% (T10) weight loss occurred, which was defined as the temperature at the beginning of thermal decomposition, and by the temperatures at the maximum rate of compound degradation (Tmax), which were determined from differential thermogravimetric analysis (DTA). The glass transition temperature (Tg) was characterized using DSC and DMTA measurements. The results obtained from thermal measurements are collected in Table 2. The TGA and DMTA graphs are shown in Figure 2.

The TGA thermographs of polythiophene confirmed the pure polymer preparation. TGA thermograph showed a two-step degradation process, where the first step of degradation visible in the peak at 73 °C corresponded with solvent evaporation, remaining after the synthesis. The second step of degradation visible on the DTA graph at 483 °C indicated thermal depolymerization of the polymer, as reported previously [25]. The T10 temperature, which represents the thermal stability of polymer, for polythiophene occurred at 315 °C. Polyamide showed a one-step degradation process; the polymer was stable until 430 °C (T10). The thermal degradation of polyamide, which occurred on the DTA graph at 467 °C, is also reported at literature [26,27].

TGA thermographs of the composites are presented in Figure 2 and indicate that the composites had better thermal stability (verify by Tmax) than polythiophene but were worse than polyamide used in their preparation. The results confirmed that PT content determined the thermal stability of composites, and they indicate substrate consequences in the context of the manufacturing method. Increase in PT concentration elevated the Tmax of composites. All composites showed a two-step degradation process. The first weight loss for composites was attributed to the evaporation of solvent and was visible for all composites in the temperature range between 43 and 78 °C. The second weight loss was attributed to degradation of composite and occurred between 462 and 475 °C. The second weight loss for PAPT1Al, which contains the biggest amount PT, occurred at 475 °C, for PAPT2Al at 472 °C and for PAPT3Al and PAPT4Al at 469 °C. For PAPT7Al, which contained the smallest amount of PT, the weight loss occurred at 464 °C. The second weight losses for composites spun with the silver substrate were visible as follows: PAPT1Ag at 475 °C, PAPT2Ag at 473 °C, PAPT3Ag at 469 °C, PAPT4Ag at 467 °C and PAPT7Ag at 462 °C. In general, the thermographs showed that the composite with the most PT had the highest degradation temperature and showed the best thermal stability. Composites with 5% PT showed the same temperature for the second thermal degradation step both for the aluminum and silver substrate. The composites containing 0.5% PT (PAPT5Al and PAPT5Ag) showed thermal degradation peaks at 467 and 463 °C, respectively. The composites containing 0.05% PT (PAPT5Al and PAPT5Ag) showed thermal degradation peaks at 464 and 462 °C, respectively. The differences in degradation temperature between composites prepared on aluminum and silver substrates were negligible. The biggest difference was 4 °C between PAPT5Al and PAPT5Ag composites. The T5 temperature of all composites was in the range of 226 to 285 °C. The T10 temperature of composites was in the range of 409–416 °C. The T5 and T10 temperatures changed with PT concentration in the composite. The T5 and T10 temperatures decreased with increasing PT content in composites. The T5 temperatures were 226 °C for PAPT1Al and 228 °C for PAPT1Ag, and the T10 temperatures were 409 °C for PAPT1Al and 410 °C for PAPT1Ag. The T5 and T10 temperatures were similar for both composites made on aluminum and silver substrates. The final residue after thermal degradation was in the range of 1.9% and 0.1% of the initial weight and was the biggest for composites containing 5% PT.

Glass transition temperatures of polymers and composites are shown in Table 2. The Tg of polythiophene occurred at 60 °C and of polyamide at 52 °C. Tm for polythiophene occurred at 204 °C and for polyamide at 228 °C. The Tg and Tm for polymers were determined by the DSC method. The results correspond with the literature [28]. Composite glass and melting transition temperatures were determined by DSC and DMTA methods. The DSC thermographs of composites are showed at Figure 3 and exhibit an endothermic peak in the range of 30–90 °C, which occurred during the first scan due to solvent evaporation. The solvent evaporation was also observed in the TGA thermographs. The second DSC peak of composites appeared in the range of 53–61 °C and was related with the glass transition temperature of composites. The third DSC peak appeared in the range of 215–222 °C and was related with melting temperature of composites. Tg obtained for composites with the DMTA method appeared in the range of 48–66 °C. The DMTA glass transition temperature values were higher than the values obtained with the DSC method. Differences between the two methods were common, as the exact position of Tg depends on the frequency used in DMTA, whereas in DSC it depends on the heating rate used [29]. In this study, the DSC heating rate was 20 °C/min, whereas the DMTA heating rate was 3 °C/min. Tg measured with DSC for composites were as follows: 61 °C for PAPT1Al, 58 °C for PAPT2Al, 57 °C for PAPT3Al and 55 °C for PAPT4Al. For composites prepared on silver substrates: 58 °C for PAPT1Ag, 58 °C for PAPT2Ag, 57 °C for PAPT3Ag and 56 °C for PAPT4Ag. The differences between composites containing various amounts of PT were clearly seen. The biggest Tg was found for the composite with 5% PT and the smallest with 0.05% PT. Such dependency was real both for composites made on aluminum and silver substrates. Composite Tg values measured with DMTA were 60 °C for PAPT1Al, 58 °C for PAPT2Al, 58 °C for PAPT3Al and 57 °C for PAPT4Al. As we could see, the glass temperature was related with the polythiophene content in the composite and was higher with higher PT concentration in composite. For composites prepared with silver substrates the values were 66 °C for PAPT1Ag, 62 °C for PAPT2Ag, 59 °C for PAPT3Ag and 58 °C for PAPT4Ag. The dependency of Pt amount in the composite was similar for aluminum substrate composites: the higher the concentration of PT in the composite, the higher the Tg value. The difference between Tg for PAPT1Al obtained with the DSC and DMTA methods was 1 °C, for PAPT2Al 0 °C and for PAPT3Al 1 °C. The melting transition temperature of polymers and composites is shown in Table 2. Tm obtained for composites with the DSC method appeared in the range of 215–222 °C. Tm values measured for composites were as follows: 218 °C for PAPT1Al, 220 °C for PAPT2Al, 220 °C for PAPT3Al and 218 °C for PAPT4Al. For composites prepared on silver substrates: 222 °C for PAPT1Ag, 221 °C for PAPT2Ag, 220 °C for PAPT3Ag and 219 °C for PAPT4Ag. The biggest Tm was found for the composite containing 5% PT and the smallest with 0.05% PT. Such dependency was real both for composites made on aluminum and silver substrates. The dependency of PT amount in the composite was similar as for dependency of glass transition temperature for composites. The melting temperature decreased with decreasing PT concentration.

The mechanical characterization of composites was performed in dual cantilever mode in the temperature range of −80 °C to 220 °C at a heating rate of 3 °C/min and frequency of 1 Hz. The dimensions of the samples were 35.0 × 10.2 × 1.0 mm. Figure 2 shows the variations in the loss modulus, storage modulus and tangent δ of the representative material.

Table 3 and Figure 4 show the relationship of the storage and loss modulus of the composites versus the temperature. There were two peaks visible for storage, loss modulus and tangent delta (tan δ). The storage modulus peaks were visible between 20 and 70 °C for region (α) and between −10 and 20 °C for region (β). The α relaxation is the mechanical manifestation of the glass transition temperature. The β relaxation was almost the same for all composites independently from PT concentration. This temperature transition is known to be related to thiophene ring motions within the chains, which were very similar for the polyamide and polythiophene in the composites. Increasing the temperature led to decreasing the composite storage modulus. Similarly to the storage modulus above the peak maximum, the loss modulus started to decrease while the temperature of the process increased. The tangent δ values for samples were represented by one peak between 20 and 75 °C, which is in agreement with others for glass transition temperature of tested composites. The values of DMTA measurements agreed with literature studies [30]. The DMTA results were dependent on the substrate used and PT content. E′ values are listed above, for PAPT1Al as well as PAPT1Ag, which at −10 °C were 3.2 and 2.9 GPa, respectively; at 20 °C, 4.0 and 3.5 GPa for PAPT1Al and PAPT1Ag, respectively; and for 70 °C the values were 2.0 and 1.6 GPa. E′ at 20 °C for PAPT2Al was equal to 3.8 GPa and for PAPT2Ag 3.5 GPa, for PAPT3Al 3.5 GPa and for PAPT3Ag 2.6 GPa, for PAPT4Al 2.5 GPa and for PAPT4Ag 1.8 GPa, for PAPT5Al 1.8 GPa and for PAPT5Ag 1.5 GPa, for PAPT6Al 1.5 GPa and for PAPT6Ag 1.5 GPa, for PAPT7Al 1.4 GPa and for PAPT7Ag 1.4 GPa. Comparing the E′ values at one temperature, one can say that the silver substrate resulted in lower E′ values. The results were lower in range of 0–0.9 GPa. The differences were clearly seen for composites with varied PT concentrations. The results taken from one substrate and different content of PT show that increasing the PT concentration in the composite resulted in higher E′. This indicates that the composite with 5% of PT was the most stiff. The same dependency was seen with E″ values, which are listed in Table 3. Increasing the concentration of PT in composite resulted in higher E″ values. This means that the composites with higher PT amounts showed more material resistance.

The results of conductivity based on resistivity measurements are given in Table 4. Resistivity was measured up to 200 ohm, and higher results were noted as no results. The measurements were repeated 15 times on each sample, and the electrodes were applied in the same distance of 1cm. The conductivity of the prepared composites was dependent on the PT content and substrate used for preparing the composite. The composite mats prepared on aluminum exhibited conductivity for samples with 5, 3 and 2% PT content, whereas conductivity was not detected in the case of silver substrate manufacturing. The highest conductivity, as could be expected, was found for the composite with highest content of PT (5.0%). The literature evidence suggests that such a conductivity value is promising for further investigations, that is, for this material’s use in electrode preparation for DSSCs [31].

The particle size of PT, employed in composites manufacturing, was measured with dynamic light scattering in a formic acid solution at 25 °C. The results showed that composites had similar particle diameters in solution. The maximum difference was 3 nm.

The morphological results are shown in Table 4, Figure 5 and Figure 6. The SEM images presented in Figure 6 shows the surfaces of the polymer composites. The fiber diameter was averaged for 100 fibers from at least five micrographs. The composites had a three-dimensional network structure composed of randomly oriented polymer fibers. The composites prepared by electrospinning were analyzed via SEM and showed yarn sizes between 149 nm and 185 nm, depending on PT content. The composite with 5% PT content exhibited the biggest yarn sizes of 184 and 185 nm for PAPT1Al and PAPT1Ag, respectively. The composites with 3% of PT were as follows: PAPT2Al showed the yarn size of 179 nm and PAPT2Ag 182 nm. Composite PAPT3Al exhibited a yarn size of 171 nm and composite PAPT3Ag 170 nm. Figure 5 shows the dependence between PT concentration in composite and yarn size. The analysis of results indicated that the yarn size decreased within decreasing PT concentration in composites. For example, the PAPT1Al composite showed a yarn size of 184 nm and PAPT7Al 152 nm, and the composite PAPT1Ag showed a yarn size of 185 nm and PAPT7Ag 149 nm.

The yarn size was related to the conductivity of the composites; the higher the conductivity of the material, the greater the yarn size. Increasing the polyaniline content in the composite increased the formation of small agglomerates on fiber surfaces, which can be explained by the increasing conductivity of the composites. The yarn sizes were in agreement with the particle sizes determined by DLS in water.

## 3. Materials and Methods

### 3.1. Materials

Thiophene (≥99%, Sigma-Aldrich, Saint Louis, MI, USA), anhydrous ferric chloride (≥99%, Sigma-Aldrich, Saint Louis, MI, USA), polyamide 6 (Sigma-Aldrich, Saint Louis, MI, USA), formic acid (97%, Sigma-Aldrich, Saint Louis, MI, USA) and methanol (≥99%, Sigma-Aldrich, Saint Louis, MI, USA) were used as received. Chloroform was dried over phosphorus pentoxide under reflux and used immediately after distillation.

### 3.2. Synthesis

The synthesis was carried out via oxidative polymerization with anhydrous ferric chloride in dry chloroform. Polymer was synthesized at room temperature (20 °C) under argon atmosphere. Thiophene (12 mmol) was added into a dispersion of ferric chloride (48 mmol) of chloroform (150 mL). After 24 h, the polymer was precipitated in methanol and filtered on filter paper. The obtained polymer was dissolved in chloroform and re-precipitated in methanol to be purified and was dried under reduced pressure. The reaction yield was 84%.

### 3.3. Electrospinning

The solutions of composites were made by dissolving polythiophenepolythiophene (PT) (1.53 g) in polyamide (PA) (6.70 g) solution in formic acid (30.50 g; 25.00 mL, 18%). The solutions were mixed 2 h before the electrospinning process. The solution flow rate was 0.20 mL/h, the needle diameter was 0.5 mm and the voltage was 25 kV. The composites were prepared with 0.05%, 0.5%, 2% and 5% mass percent of polythiophene in solution by diluting the base solution. The solutions were electrospun on aluminum (Al) and silver (Ag) round substrates. Table 1 represents the preparation and names of samples.

### 3.4. Methods

Thermogravimetric analysis (TGA) measurements were performed using a TGA 55 (TA Instruments, New Castle, DE, USA) device. Experiments were carried out in a nitrogen stream (20 mL/min) with a scanning rate of 10 °C/min in the temperature range of 30–900 °C. Differential scanning calorimetry (DSC) measurements were performed using a Q2000 calorimeter (TA Instruments, New Castle, DE, USA) in a nitrogen stream at a scanning rate of 20 °C/min. Samples were analyzed in aluminum pans in the temperature range of −80 to 300 °C. Fourier transform infrared (FTIR) measurements were performed in the range of 4000–400 cm^−1^ with potassium bromide (KBr)-pressed pellets using a Spectrum One instrument (Perkin Elmer, New Castle, DE, USA). The samples were measured at room temperature, and pellets were prepared by mixing 10.0 mg of polymer with 100.0 mg of KBr. The sample pellets were prepared by applying high pressure to a polymer sample with KBr. Scanning electron microscopy (SEM) measurements were performed using a scanning electron microscope (Quanta 250 FEG, FEI Company, New Castle, DE, USA) operating with an acceleration voltage of 10 kV under low vacuum (80 Pa). Electron micrographs were obtained from secondary electrons collected by a large-field detector (LFD). The samples were stuck to microscopic stubs by double-sided adhesive carbon tape. Micrograph analysis was carried out using ImageJ software. Conductivity measurements were made using a two-point probe conductor using a Keithley 2400 multimeter. Sizes were determined with a Litesizer 500 (Anton Paar GmbH, Graz, Austria) equipped with a 658 nm laser. The measurements were carried out at 90° in polystyrene cuvettes. Measurements were conducted 5 times at 25 °C for 60 s, with equilibration periods of 3 min. Dynamic mechanical measurements were carried out on a DMTA Q800 (TA Instruments, New Castle, DE, USA) analyzer. DMTA allowed us to investigate the mechanical property behaviors related to Brownian motion of the polymer chains, such as variation in the mechanical modulus versus temperature. The mechanical dissipation factor, tan δ, as a measure of the deformational energy dissipated as heat during each cycle, is given by tan δ = E″/E′. The storage modulus, loss modulus and tangent δ of the samples were determined at 1 Hz and from −80 °C to 220 °C at a heating rate of 3 °C/min. The sample dimensions were 15.0 mm × 4.0 mm in compression mode. TGA, DSC, IR as well as DMTA analyses were carried out using OriginPro 2019b.

## 4. Conclusions

Investigation of parameters of composites prepared using polythiophene and polyamide indicate the possible application of these materials in construction of DSSC and others. The mechanical parameters have to be taken into consideration when controlling the conductivity in manufacturing.

## Figures and Tables

**Figure 1 molecules-26-02476-f001:**
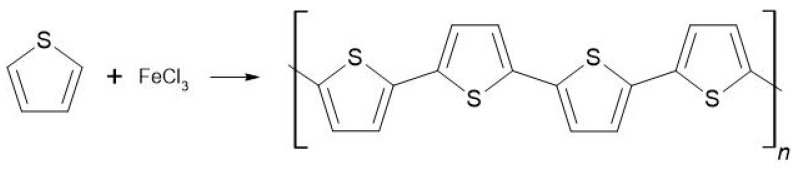
Scheme of oxidative polymerization of thiophene.

**Figure 2 molecules-26-02476-f002:**
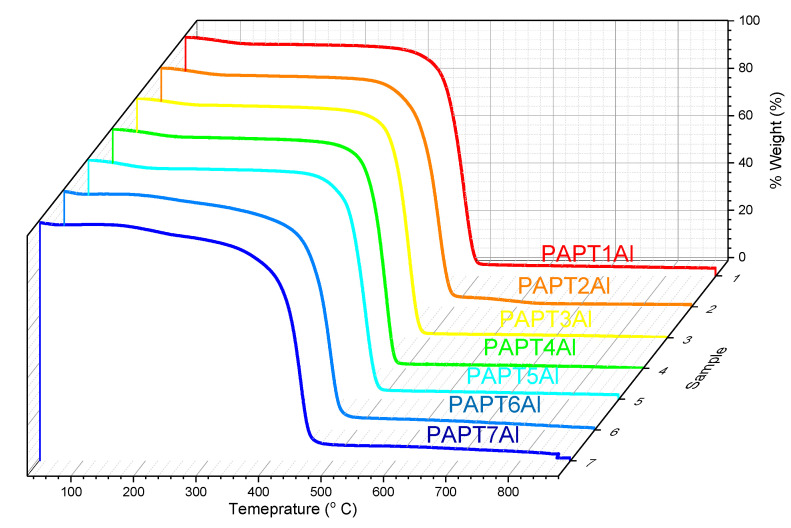
Thermogravimetric thermographs of composites.

**Figure 3 molecules-26-02476-f003:**
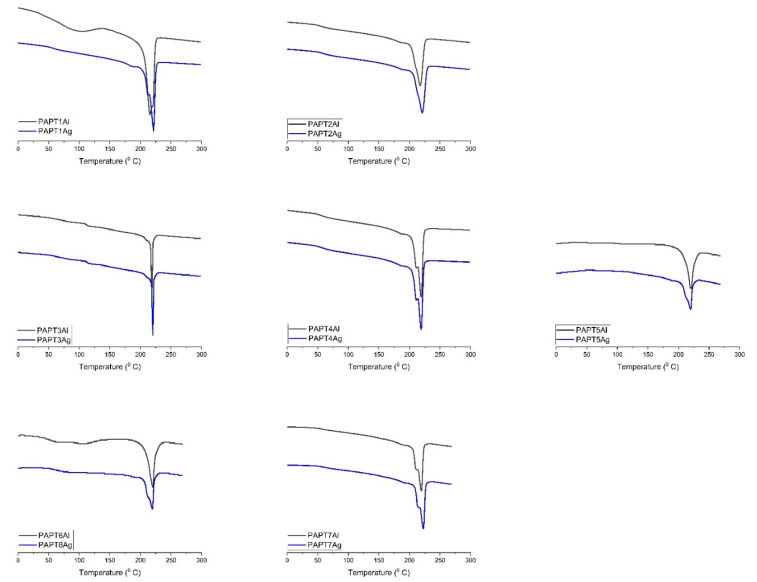
Differential scanning calorimetry of composites.

**Figure 4 molecules-26-02476-f004:**
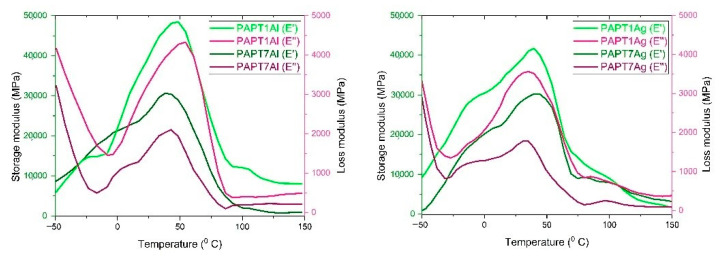
Dynamic mechanical thermal analysis of PAPT1Al, PAPT1Ag, PAPT7Al and PAPT7Ag composites.

**Figure 5 molecules-26-02476-f005:**
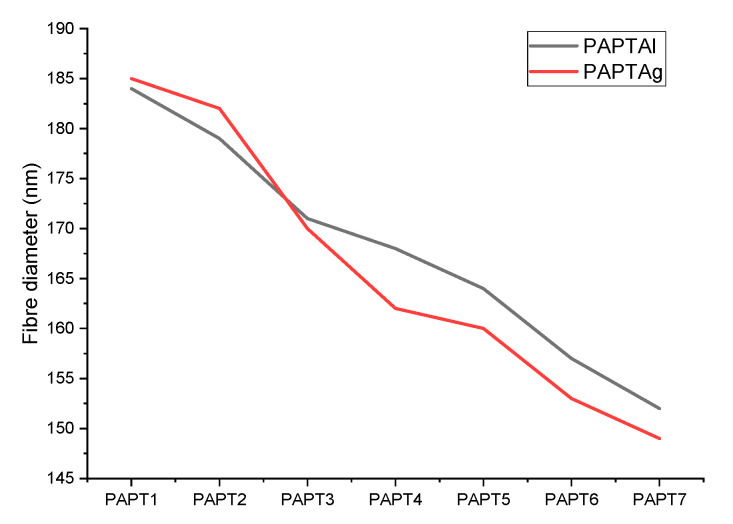
The fiber diameter of composites measured by SEM.

**Figure 6 molecules-26-02476-f006:**
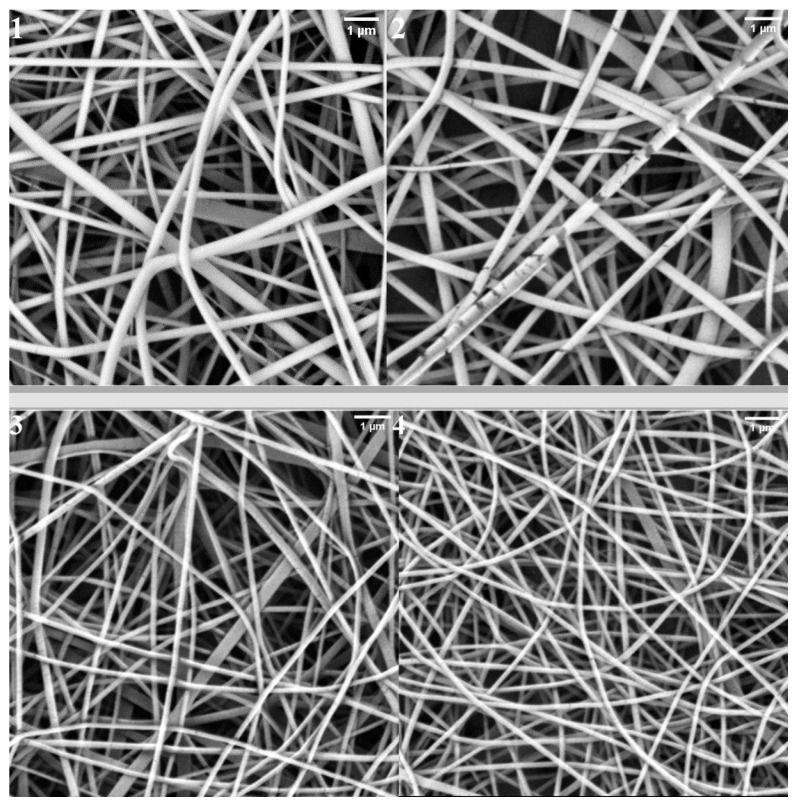
SEM micrographs of PAPT1Al (**1**), PAPT7Al (**2**), PAPT1Ag (**3**) and PAPT7Ag (**4**) composites.

**Table 1 molecules-26-02476-t001:** Data taken to prepare composite films.

Name	Substrate	Polythiophene (g)	Polythiophene (%)
PAPT1Al	aluminum	1.53	5%
PAPT2Al		0.92	3%
PAPT3Al		0.61	2%
PAPT4Al		0.26	0.85%
PAPT5Al		0.153	0.5%
PAPT6Al		0.0765	0.25%
PAPT7Al		0.0153	0.05%
PAPT1Ag	silver	1.53	5%
PAPT2Ag		0.92	3%
PAPT3Ag		0.61	2%
PAPT4Ag		0.26	0.85%
PAPT5Ag		0.153	0.5%
PAPT6Ag		0.0765	0.25
PAPT7Ag		0.0153	0.05%

**Table 2 molecules-26-02476-t002:** Thermal properties of starting polymers and conductive composites.

Sample	T_5_ (°C)	T_10_ (°C)	Residue at 900 °C (%)	T_max_ (°C)	T_gDSC_ (°C)	T_mDSC_ (°C)	T_gDMTA_ (°C)
Polythiophene	283	315	2.6	73; 483	60	204	-
Polyamide	325	430	1.8	461	52	228	-
PAPT1Al	226	410	1.9	50; 475	61	218	60
PAPT2Al	254	414	0.6	76; 472	58	220	58
PAPT3Al	266	414	0.1	73; 469	57	218	58
PAPT4Al	273	415	0.6	72; 469	55	218	57
PAPT5Al	272	416	0.3	78; 467	55	217	54
PAPT6Al	270	415	0.3	65; 465	54	217	54
PAPT7Al	285	419	0.4	75; 464	54	215	48
PAPT1Ag	228	409	1.4	45; 475	58	222	66
PAPT2Ag	240	410	0.5	78; 473	58	221	62
PAPT3Ag	267	412	0.2	43; 469	57	220	59
PAPT4Ag	271	414	0.3	73; 467	56	219	58
PAPT5Ag	272	414	0.4	47; 463	54	217	54
PAPT6Ag	275	415	0.1	48; 463	54	216	51
PAPT7Ag	276	415	0.2	60; 462	53	216	49

T5 and T10 represent the temperatures of 5 and 10% weight loss. Tmax is the temperature of the maximum decomposition rate as determined by DTA. TgDSC is the glass transition temperature by DSC, and TgDMTA is the glass transition temperature by DMTA. TmDSC is the melting temperature by DSC.

**Table 3 molecules-26-02476-t003:** Storage and loss modulus versus temperature of composites.

Name	E′ (GPa)			E″ (GPa)		
	−10 °C	20 °C	70 °C	−10 °C	20 °C	70 °C
PAPT1Al	3.2	4.0	2.0	1.7	3.1	0.7
PAPT2Al	3.1	3.8	2.1	1.6	2.3	0.2
PAPT3Al	2.9	3.5	1.8	1.5	1.6	0.1
PAPT4Al	2.3	2.5	1.3	1.2	1.3	0.1
PAPT5Al	2.2	1.8	1.0	0.9	1.1	0.3
PAPT6Al	2.0	1.5	1.0	0.8	1.1	0.2
PAPT7Al	2.0	1.4	1.0	0.8	1.0	0.2
PAPT1Ag	2.9	3.5	1.6	1.8	3.2	1.0
PAPT2Ag	2.8	3.5	1.4	1.7	3.0	0.8
PAPT3Ag	2.2	2.6	1.3	1.6	2.8	0.3
PAPT4Ag	2.2	1.8	1.2	1.6	1.9	0.1
PAPT5Ag	2.0	1.5	1.0	1.4	1.6	0.1
PAPT6Ag	2.0	1.4	1.0	1.2	1.4	0.2
PAPT7Ag	1.9	1.4	0.9	1.2	1.3	0.2

**Table 4 molecules-26-02476-t004:** Electric conductivity and fiber size of composites.

Sample	Conductivity (mS/cm)	Particle Size in Solution (nm)	Yarn Size (nm)
PAPT1Al	29.4	23	184
PAPT2Al	11.9	23	179
PAPT3Al	8.7	23	171
PAPT4Al		23	168
PAPT5Al		23	164
PAPT6Al		23	157
PAPT7Al		23	152
PAPT1Ag		23	185
PAPT2Ag		23	182
PAPT3Ag		23	170
PAPT4Ag		23	162
PAPT5Ag		23	160
PAPT6Ag		23	153
PAPT7Ag		23	149

## Data Availability

The data presented in this study are available in this article.

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
