# Peer review of "Effect of Polythiophene Content on Thermomechanical Properties of Electroconductive Composites"

_molecules, 2021, doi:10.3390/molecules26092476_

Round 1

Reviewer 1 Report

The manuscript is devoted to reveal the impact of the polythiophene content on properties of electrospun fibers by several technique like FT-IR, TGA, DSC, DMTA and SEM.

I recommend to publish this manuscript if the following points are fulfilled:

1) In Figure 3 are presented the DSC of composites with Al substrates but for comparative purposes I recommended to show the DSC of composites with silver substrates.

2) Line 207 change Figure 2 with Figure 4.

3) Line 210 change Figure 3 with Figure 4.

4) In Figure 4 highlights by comparison the DSC for PAPT1Al vs. PAPT7Al and PAPT1Ag vs. PAPT7Ag.

5) Figure 5 it is not explained in the text.

6) Show the SEM micrographs for PAPT1Al vs. PAPT7Al and PAPT1Ag vs. PAPT7Ag.

Author Response

Response to review:

1) In Figure 3 are presented the DSC of composites with Al substrates but for comparative purposes I recommended to show the DSC of composites with silver substrates.

The authors attached the DSC of composites with silver susbtrates to the manuscript.

2) Line 207 change Figure 2 with Figure 4.

The authors changed Figure 2 with Figure 4 at line 207.

3) Line 210 change Figure 3 with Figure 4.

The authors changed Figure 3 aith Figure 4 at line 210.

4) In Figure 4 highlights by comparison the DSC for PAPT1Al vs. PAPT7Al and PAPT1Ag vs. PAPT7Ag.

The graph has been changed, authors added DMTA of PAPT1Al vs. PAPT7Al and PAPT1Ag vs. PAPT7Ag figures.

5) Figure 5 it is not explained in the text.

The authors added explanation of Figure 5 to the text.

6) Show the SEM micrographs for PAPT1Al vs. PAPT7Al and PAPT1Ag vs. PAPT7Ag.

The authors added indicated micrographs to Figure 6.

Reviewer 2 Report

The authors declare that the present work is aimed at proposing a composite material based on conductive polymers, reinforced with a microfiber base layer obtained by electrospinnig, applicable to the manufacture of dye-sensitized solar cells (DSSC). Specifically, we speak of a composite material of polythiophene (PT) and polyamide (PA6). But instead, in the title it is stated that the compound is polyaniline (be careful with this formal error, which is in a very prominent point!).

In the work, a method of preparing a very complete set of composites with different PT content is exposed. The preparation techniques are well detailed, and their thermal, viscoelastic, micromorphological and electrical properties are efficiently evaluated and defined, with appropriate techniques, and with a well-defined characterization testing methodology. The preparation and characterization work of the indicated properties, in my opinion, is very complete. The writing is clear and the manuscript in general is well structured.

But I do not see in any section of the work an essay that directly characterizes the material prepared as suitable to give an efficient response in a photovoltaic solar cell. Or at least, some spectrophotometric determination of some property directly related to the photoelectric properties of the material.

The authors must provide this information in some way, as a requirement for the publication of this interesting work in this journal.

Author Response

Response to review:

The title in the manuscript was introduced by mistake. The authors has changed the title of manuscript.

We have not tested yet our materials in a photovoltaic solar cell, and on this stage of our work any spectrophotometric investigations have not been carried out. We would like to stressed that this work presents the results of the basic research aimed to synthesis and characterization of conductive polymer composites including its thermal, thermomechanical and conductivity properties together with morphology investigations. One of the idea of study of such materials, as was mentioned in Introduction, can be its potential utilization for PV device preparation mainly dye-sensitized (DSSCs) or perovskite (PSCs) solar cells. It seems to be interesting issue because it was found that various types of polymers including semiconductor, or a classical insulator used to dope into a perovskite film or for passivation of perovskite layer results in increased device performance [RSC Adv., 2015, 5, 775 , Adv. Energy Mater. 2017, 1701757; NATURE COMMUNICATIONS  (2019) 10:520; Science 371, 390–395 (2021); Sci. Adv. 2017;3: e1700106]. Moreover, conductive polymers or its composites are investigated as perspective alternative for Pt counter electrodes [Organic Electronics 29 (2016) 107-113; Journal of Power Sources 284 (2015) 489-496; Nanoscale, 2013, 5, 7838–7843; J Polym Res (2016) 23: 192; European Polymer Journal 66 (2015) 207–227] or for liquid electrolytes in DSSCs [Current Applied Physics 18 (2018) 619–625]. The preparation of conductive polymer composites may improve the drawback of a neat conducting polymers, which are usually mechanically weak and insoluble. The next step of our investigations, which is out of the scope of the present work, it will be attempts to applied the selected materials for DSSC fabrication.  

Round 2

Reviewer 2 Report

Once the authors have corrected the formal error in the title of the work, and having considered reasonable the scientific-technical arguments presented by the authors to justify the absence of specific experimentation of the photovoltaic effect of the material they have prepared, I have decided to consider this this one suitable. work for publication in this journal in its current form.